# NIR-Triggered Generation of Reactive Oxygen Species and Photodynamic Therapy Based on Mesoporous Silica-Coated LiYF_4_ Upconverting Nanoparticles

**DOI:** 10.3390/ijms23158757

**Published:** 2022-08-06

**Authors:** Tsung-Han Ho, Chien-Hsin Yang, Zheng-En Jiang, Hung-Yin Lin, Yih-Fung Chen, Tzong-Liu Wang

**Affiliations:** 1Department of Chemical and Materials Engineering, National Kaohsiung University of Science and Technology, Kaohsiung 807, Taiwan; 2Department of Chemical and Materials Engineering, National University of Kaohsiung, Kaohsiung 811, Taiwan; 3Graduate Institute of Natural Products, Kaohsiung Medical University, Kaohsiung 807, Taiwan

**Keywords:** mesoporous silica, reactive oxygen species, photodynamic therapy, upconversion nanoparticles, photosensitizers, fluorescence resonance energy transfer

## Abstract

To date, the increase in reactive oxygen species (ROS) production for effectual photodynamic therapy (PDT) treatment still remains challenging. In this study, a facile and effective approach is utilized to coat mesoporous silica (mSiO_2_) shell on the ligand-free upconversion nanoparticles (UCNPs) based on the LiYF_4_ host material. Two kinds of mesoporous silica-coated UCNPs (UCNP@mSiO_2_) that display green emission (doped with Ho^3+^) and red emission (doped with Er^3+^), respectively, were successfully synthesized and well characterized. Three photosensitizers (PSs), merocyanine 540 (MC 540), rose bengal (RB), and chlorin e6 (Ce6), with the function of absorption of green or red emission, were selected and loaded into the mSiO_2_ shell of both UCNP@mSiO_2_ nanomaterials. A comprehensive study for the three UCNP@mSiO_2_/PS donor/acceptor pairs was performed to investigate the efficacy of fluorescence resonance energy transfer (FRET), ROS generation, and in vitro PDT using a MCF-7 cell line. ROS generation detection showed that as compared to the oleate-capped and ligand-free UCNP/PS pairs, the UCNP@mSiO_2_/PS nanocarrier system demonstrated more pronounced ROS generation due to the UCNP@mSiO_2_ nanoparticles in close vicinity to PS molecules and a higher loading capacity of the photosensitizer. As a result, the three LiYF_4_ UCNP@mSiO_2_/PS nanoplatforms displayed more prominent therapeutic efficacies in PDT by using in vitro cytotoxicity tests.

## 1. Introduction 

In recent decades, the development of nanotechnology resulted in the design of various nanomaterials for biological and therapeutic applications such as gene therapy, tissue engineering, drug delivery, molecular imaging, etc. [1,2,3,4]. In comparison with conventional therapies, nanomaterials are more efficient for cancer therapy because they can be designed appropriately to target affected tumor sites and discriminatorily distribute their loads with fewer side effects. Among the various nanomaterials, mesoporous silica nanoparticles (MSNs) have emerged as favorable inorganic material platforms for biomedical applications since 2001 [5,6,7]. Due to their large surface areas, tunable sizes, high loading capacity, thermal and photostability, facile modification and considerable biocompatibility, mesoporous silica (mSiO_2_) functionalized nanocomposites have been widely used in many biomedical applications such as drug delivery, photodynamic therapy (PDT), etc. [8,9,10,11,12,13,14,15]. Compared to other kind of nanoparticles, the unique properties of these inorganic nanomaterials allowed for building efficient complex nanostructures. Consequently, mesoporous silica nanoparticles such as MCM-41, MCM-48, SBA-15, and core-shell and hollow MSNs have been used in drug delivery due to their distinctive properties [16,17,18,19]. 

Meanwhile, there has been an awareness of the medical potential of PDT for more than 25 years. PDT pertaining to the photochemical reactions of photosensitizers (PSs) is a treatment approach utilizing light irradiation at proper wavelengths. In a PDT treatment, the photosensitizers are stimulated and produce reactive oxygen species (ROS) to give rise to oxidative stress and cellular damages [20]. Compared to the conventional cancer therapy, PDT has several advantages including precise tumor targeting and negligible systemic toxicity; it is less invasive than surgery, and is accessible for repeated therapies [21,22,23]. Among them, the utmost advantage of PDT can be attributed to its selective therapeutics for tumor cells by using appropriate light under specific control. However, most PS molecules need to be motivated by ultraviolet or visible light that has poor tissue-penetration depth, and thus restricts the phototherapy of large or internal tumors. This drawback can be avoided by adsorbing the photosensitizers on the surface or outer shell of upconversion nanoparticles (UCNPs). Upconversion nanoparticles can absorb two or more photons and give off the light at shorter wavelength than the excitation wavelength. Consequently, the utilization of UCNPs as nanocarriers facilitates the development of near-infrared (NIR) light-triggered PDT. This treatment is in relation to upconverted fluorescence emissions of UCNPs and fluorescence resonance energy transfer (FRET) between UCNPs and the photosensitizers. Several approaches have been employed to load the PS molecules onto the surface of UCNPs, which include coating with a mesoporous silica layer and encapsulating in a polymer shell [24,25,26,27,28]. To secure more efficient remedial effects, it is suggested to exploit intense and appropriate upconversion (UC) emission at certain wavelengths and achieve a higher loading capacity of the PS molecules. To intensify the selectivity with respect to tumor cells and the efficiency of PDT, fabricating UCNP-PS nanocarrier systems with high therapeutic efficacy by using more efficient and powerful synthetic strategies remains challenging. 

In recent years, lanthanide-doped upconversion nanoparticles (UCNPs) have gained tremendous focus for various applications such as biological imaging, therapeutics, photovoltaics, photonics, etc. [29,30,31,32,33,34]. Due to the bounteous and unique energy levels, lanthanide-doped UCNPs can display numerous distinctive characteristics including low autofluorescence, narrow bandwidths, long luminescence lifetime, less photobleaching, large anti-Stokes shift, superior photostability, and deep penetration depth for tissue [33,35,36,37,38,39,40,41,42]. The intrinsic photophysical and photochemical properties of UCNPs render them particularly applicable for phototherapy. Over the past decade, combining UCNPs and photosensitizers as theranostic platforms have been widely applied in PDT via in vitro and in vivo tests [43,44]. To endow their promising versatility for practical applications, a wide range of surface modification methods has been adopted for the design of nanocarriers with distinctive physico-chemical, toxicological, and pharmacological properties [31,45,46]. In particular, studies on light-controlled drug delivery systems based on MSNs have been executed extensively because light under specific control can provide well-regulated drug release both spatially and temporally, and thus present great potentials for further biomedical applications. Taking the advantages of UCNPs in phototherapy, the functionalization of the UCNP surface for various biomedical applications is intriguing for researchers in the clinical theranostic field. 

In a UCNP—PS nanocarrier system, UCNPs act as donors, while the PS molecules function as acceptors to generate ROS. Through the FRET process between UCNPs and photosensitizers, ROS can be generated and used in PDT [47,48,49]. Since the efficacy of PDT treatment depends on the efficiency of the corresponding ROS generation, the loading capacity of PS molecules on the surface of UCNPs and selection of the photosensitizer for effective FRET between UCNP—PS (donor-acceptor) system is of crucial importance. In this study, we report a more effective approach for PDT treatment based on the surface modification of LiYF_4_ UCNPs by using MSNs as compared to the PDT efficacy via the oleate-capped and ligand-free type UCNPs in our previous study [50]. To the best of our knowledge, it is hard to find relevant literature with regard to ROS generation and PDT treatment through the donor-acceptor FRET pair of the LiYF_4_ type UCNP@mSiO_2_ nanomaterials and the organic dye-based photosensitizers. It is suggested that the comparison of the PDT efficacy of the three kinds of UCNPs with different surface morphologies could be applied in all UCNP—PS nanocarrier systems. Both green emission UCNPs and red emission UCNPs composed of LiYF_4_:Yb^3+^_0.25_,Ho^3+^_0.01_@LiYF_4_:Yb^3+^_0.2_ and LiYF_4_:Yb^3+^_0.25_,Er^3+^_0.01_@LiYF_4_:Yb^3+^_0.2_ core/shell nanoparticles were synthesized, respectively. In order to establish an effective upconversion resonance energy transfer process, two photosensitizers, merocyanine 540 (MC 540) and rose bengal (RB), acted as the energy acceptor of green emission, and the other photosensitizer, chlorin e6 (Ce6), utilized as the energy acceptor of red emission were selected for this kind of donor—acceptor system. The in vitro cytotoxicity tests were also performed to evaluate the efficacy of the MSN-based surface modification for UCNPs in PDT.

## 2. Materials and Methods 

### 2.1. Synthesis of Mesoporous Silica-Coated LiYF_4_:Yb^3+^/Ho^3+^@LiYF_4_:Yb^3+^ and LiYF_4_:Yb^3+^/Er^3+^@ LiYF_4_:Yb^3+^ UCNPs

Two kinds of LiYF_4_ UCNPs capped with oleic acid and doped with Yb^3+^/Ho^3+^ (green emission) and Yb^3+^/Er^3+^ (red emission) ions were synthesized, respectively, in accordance with the previously reported method [51,52]. Subsequently, the removal of both oleate ligand from the core/shell type UCNPs composed of LiYF_4_:Yb^3+^_0.25_,Ho^3+^_0.01_@LiYF_4_:Yb^3+^_0.2_ and LiYF_4_:Yb^3+^_0.25_,Er^3+^_0.01_@LiYF_4_:Yb^3+^_0.2_ were carried out via the strategy proposed by the Capobianco group [53]. Upon the preparation of oleate-free UCNPs, the mesoporous silica was then coated on the surface of UCNPs (UCNP@mSiO_2_) (Figure 1). The experiment was achieved through a NaOH-based etching method. In a typical experiment, 0.25 g of hexadecyl trimethyl ammonium chloride (CTAC) and 0.025 g of ligand-free UCNPs were dispersed in a 50 mL aqueous solution under ultrasonic vibrating and then the temperature was raised to 75 °C. After one hour, 0.15 mL of tetraethyl orthosilicate (TEOS) was added dropwise and then 0.12 mL of NaOH aqueous solution was added into the flask. The resulting mixture was stirred for another 1 h. The reaction was added with an aliquot of ethanol to obtain the crude UCNP@mSiO_2_ (CTAC) via centrifugation. The crude product was redispersed in 2 wt% NaCl aqueous solution to remove CTAC and then recovered by centrifugation. This step was repeated three times to acquire the final product. The UCNP@mSiO_2_ nanomaterials with the function of green emission and red emission were prepared for this work. The final products were dispersed and stored in deionized water and ready for use. 

### 2.2. Adsorption of Photosensitizers into the Hollow Pores of UCNP@mSiO_2_


In this study, 6 mg of photosensitizers merocyanine 540 (MC 540), rose bengal (RB), and chlorin e6 (Ce6) were individually loaded into the pores of the mesoporous silica by soaking 10 mg of their corresponding UCNP@mSiO_2_ nanoparticles in 10 mL of solvent for 24 h at room temperature. Deionized (DI) water was used as the solvent of MC 540 and RB, while ethanol was used as the solvent of Ce6. The nanoparticles were then centrifuged and washed with the solvent (DI water or ethanol) three times to remove non-adsorbed molecules of the photosensitizers. The loading capacity of three photosensitizers was obtained from the calibration curve of their UV-vis absorption spectra at the corresponding peaks. The loading capacity of each photosensitizer was calculated as follow: loading capacity (%, *w*/*w*) = (weight amount of photosensitizer incorporated into nanoparticles)/(weight amount of nanoparticles) × 100%.

### 2.3. NIR-Induced Reactive Oxygen Species (ROS) Generation

The ROS production upon NIR irradiation was determined by using a 9,10-anthracenediyl-bis(methylene) dimalonic acid (ABDA) probe in the aqueous solution. Two mesoporous silica- coated UCNPs loaded with three different photosensitizers can be all designated as UCNP@mSiO_2_/PS or individually denoted as Ho^3+^-doped UCNP@mSiO_2_/MC 540, Ho^3+^-doped UCNP@mSiO_2_/RB, and Er^3+^-doped UCNP@mSiO_2_/Ce6. The above three nanohybrids were dispersed in DI water with a concentration of 1 mg/mL, while the ABDA solution was prepared in a concentration of 3 mg/mL. Then 2.5 mL of each nanohybrid aqueous solution was mixed with 0.5 mL of ABDA solution. The solutions were placed in quartz cuvettes and irradiated for 60 min using 3 W/cm^2^ 980 nm laser. The absorption intensities for the three peaks at 359 nm, 378 nm and 399 nm were recorded every 10 min interval by UV-vis spectrophotometer. ROS generation was investigated by detecting the absorptions of ABDA at the three peaks.

### 2.4. Cytotoxicity Tests of Cancer Cells

Cytotoxicity tests were measured using CCK-8 kits. The MCF-7 cells were seeded in 96-well plates. After cultivation for 24 h, 100 μL of UCNP@mSiO_2_/PS nanomaterials was added into the culture medium at different concentrations, with five parallel wells for each concentration (0, 50, 100, 250, and 500 μg/mL). After irradiation with 980 nm NIR for 5 min, the cells were then incubated for another 24 h in an incubator and added with 10 μL of CCK-8 and 90 μL of medium assay per well. After 3 h of incubation, the cell viability was obtained by an ELISA reader.

### 2.5. Characterization 

X-ray photoelectron spectroscopy (XPS) analysis was performed by a JEOL JAMP-9500F Auger electron spectrometer. Energy-dispersive X-ray spectroscopy (EDS) was conducted in a FEI Genesis XM 4i energy dispersive X-ray analysis system. Wide-angle X-ray diffraction (WAXD) analysis were performed with a Bruker D8 ADVANCE diffractometer, using Cu Kα radiation with a step size of 0.05° and a scanning speed of 4°/min. Transmission electron microscopy (TEM) data were collected with a JEOL JEM1230 transmission electron microscope. Ultraviolet-visible (UV-vis) spectroscopic analysis was implemented by a Perkin Elmer Lambda 35 UV-vis spectrophotometer. Photoluminescence (PL) spectra were recorded on a Hitachi F-7000 fluorescence spectrophotometer (Japan). Dynamic light scattering (DLS) measurements were performed using a Brookhaven Instruments 90PLUS instrument. The emission spectra of the upconversion nanoparticles were recorded upon irradiation with 980 nm NIR laser using a SDL980-LM-5000T (Shanghai Dream Lasers Technology Co., Ltd., Shanghai, China) laser diode at 3 W/cm^2^.

## 3. Results and Discussion

### 3.1. Synthesis of Mesoporous Silica-Coated UCNPs

A general strategy to achieve the NIR-triggered ROS generation in UCNPs is to modify the surface of UCNPs with the function of adsorption of photosensitizers. Since the as-synthesized UCNPs were oleate-capped and hydrophobic, the surface modification via the coating of mesoporous silica (mSiO_2_) were utilized to render them hydrophilic for the related biological applications. An overcoating of mesoporous silica on the UCNP surface has several advantages, including low cytotoxicity, high chemical stability, etc. Above all, the large surface area and pore volume of mesoporous silica secure easier adsorption as well as a higher loading capacity of various therapeutic materials. For the preparation of mesoporous silica-coated UCNPs, many studies employed the reverse micro-emulsion method to directly coat mesoporous SiO_2_ on the hydrophobic UCNPs. However, this method needs to add a surfactant in the reaction mixture. In addition, the existence of organic ligands (such as oleic acid) on the UCNP surface may hinder the process of FRET and the homogeneity of mesoporous silica layer. More importantly, the agglomeration phenomenon of UCNPs caused by the organic ligands during the mSiO_2_ coating process resulted in the lower loading capacity (%, *w*/*w*) for every gram of mSiO_2_-coated UCNP particles, as we found in some previous studies [43,48]. Consequently, the preparation of mesoporous silica-coated UCNPs in this study was carried out through a modified approach unlike most reported methods. The synthetic route of UCNP@mSiO_2_ was achieved by coating the silica layer on the surface of oleate-free UCNPs with the addition of CTAC template and NaOH etchant to form the mesoporous silica shell. In this work, two kinds of mesoporous silica-coated UCNPs with the emission in the green (doped with Yb^3+^/Ho^3+^) and red (doped with Yb^3+^/Er^3+^) regions were prepared. The green emission UCNP is adopted as the donor for the photosensitizers MC 540 and rose bengal because both PSs have strong absorption in the green region, while the red emission UCNP is selected for the photosensitizer Ce6 due to its strong absorption band in the red region. Firstly, both oleate-free type UCNPs, green emission of LiYF_4_:Yb^3+^_0.25_,Ho^3+^_0.01_@LiYF_4_:Yb^3+^_0.2_ nanoparticles and red emission of LiYF_4_:Yb^3+^_0.25_,Er^3+^_0.01_@LiYF_4_:Yb^3+^_0.2_ nanoparticles were synthesized according to the published article [50]. The coating of silica on the surface of UCNPs was then carried out by adding TEOS and using CTAC as the template to form the silica-coated UCNPs (UCNP@SiO_2_). Afterwards, the mesoporous silica coating was achieved by the NaOH etching and ion exchange with NaCl to obtain the mesoporous silica-coated UCNPs (UCNP@mSiO_2_). After coating with the mesoporous silica, it is apparent that both mSiO_2_-coated UCNPs displayed excellent dispersibility in aqueous solutions. 

### 3.2. Structural Characterization of the m-SiO_2_ Coated UCNPs

X-ray diffraction (XRD) analysis was used to clarify the phase structures of both Ho^3+^-doped and Er^3+^-doped oleate-capped core/shell UCNPs and mesoporous silica-coated UCNPs. As regards both mSiO_2_-coated UCNPs, the XRD patterns shown in Figure 1a,b are similar to those of oleate-capped UCNPs, which are indexed as the standard data (JCPDS No. 17-0874) of the tetragonal LiYF_4_ crystal. It is obvious that the coating of a thin layer mesoporous silica did not affect the crystal structure of the synthesized UCNPs. It should be noted that the WAXD diffractograms of both ligand-free UCNPs are omitted here due to the fact that they have the identical structures with those of oleate-capped UCNPs [51]. 

The surface morphology of both mSiO_2_-coated UCNPs is observed with TEM and shown in Figure 2 and Figure 3. As evident from Figure 2 and Figure 3, a thin layer of mesoporous silica is present on the surface of LiYF_4_:Yb^3+^_0.25_,Ho^3+^_0.01_@LiYF_4_:Yb^3+^_0.2_ and LiYF_4_:Yb^3+^_0.25_,Er^3+^_0.01_@LiYF_4_:Yb^3+^_0.2_ UCNPs. Compared to the ligand-free UCNPs, a homogeneous coating of mSiO_2_ layer on the UCNP surface appears both in the long axis and short axis of the tetragonal crystals. 

As seen in Figure 2 and Figure 3, with respect to both green emission and red emission UCNPs, it is evident that the particle sizes increased after the mesoporous silica layer was coated. For easier comparison, a representative particle of each oleate-free and mesoporous silica-coated UCNP sample was chosen and the size was labeled, as shown in Figure 2 and Figure 3. The long axis and short axis for the ligand-free Ho^3+^-doped UCNPs and the Er^3+^-doped UCNPs are 159.64 nm and 69.13 nm, and 152.19 nm and 71.62 nm, respectively. The calculated aspect ratios (major axis/minor axis) are approximately 2.31 and 2.12. After overcoating the mSiO_2_ layer, it is apparent that both ligand-free UCNPs display the increases in size in both long axis and short axis. The long axis and short axis for the Ho^3+^-doped and Er^3+^-doped nanoparticles are 185.24 nm and 108.40 nm and 177.11 nm and 102.78 nm, respectively.

Specific surface area measurements of both mSiO_2_-coated UCNPs were performed by the Brunauer-Emmett-Teller (BET) method through the adsorption of nitrogen gas. According to the IUPAC classification scheme for mesoporous materials, both N_2_ adsorption/desorption isotherms of Ho^3+^-doped and Er^3+^-doped UCNP@mSiO_2_ (Appendix A) indicate that the isotherms of UCNP@mSiO_2_ nanomaterials exhibit a type-IV with a hysteresis loop. The surface areas for both UCNP@mSiO_2_ are determined to be 139.1 and 232.8 m^2^/g, respectively, using the BET method. The average pore radii (Appendix A) are determined to be 13.7 and 13.0 nm, correspondingly, suggesting that both as-synthesized UCNP@mSiO_2_ nanocomposites are porous structures and can be used for loading photosensitizers or drugs.

Dynamic light-scattering (DLS) analyses were carried out to further confirm the change in the particle size after the surface of UCNPs was coated with mSiO_2_. As shown in Appendix A, for the green emission LiYF_4_:Yb^3+^_0.25_/Ho^3+^_0.01_@ LiYF_4_:Yb^3+^_0.2_ UCNPs, the average particle size changed from 145.6 nm (ligand-free) to 183.4 nm after covering with mSiO_2_. Similarly, as shown in Appendix A, the average particle size of the red emission LiYF_4_:Yb^3+^_0.25_/Er^3+^_0.01_@ LiYF_4_:Yb^3+^_0.2_ UCNPs increased from 141.2 nm (ligand-free) to 174.3 nm (mSiO_2_-coated). The results of DLS data testified the successful coating of mesoporous silica on the surface of oleate-free UCNPs. Appendix A depicts that the zeta potentials for Ho^3+^-doped and Er^3+^-doped UCNP@mSiO_2_ are around −19.05 and −23.29 mV, respectively.

The successful coating of an mSiO_2_ layer on the ligand-free UCNPs was further verified by the XPS and EDS data, as shown in Figure 4. The XPS spectra in Figure 4a show the Ho 3d band for the Ho^3+^-doped ligand-free UCNPs and mSiO_2_-coated UCNPs. Similarly, the existence of Er 3d band in Figure 4b confirms the structure of both Er^3+^-doped ligand-free and mSiO_2_-coated UCNPs. As compared to both ligand-free UCNPs, the presence of intense peaks corresponding to Si 2p and O 1s indicates the successful coating of mSiO_2_ on the UCNP surface. Nevertheless, both Si 2p and O 1s peaks are also perceptible in both ligand-free type UCNPs because the XPS measurements were performed on the glass substrates. The EDS spectra exhibit the existence of the elements of Y, Yb, F, Si, and O in the Ho^3+^-doped UCNP@mSiO_2_ and the Er^3+^-doped UCNP@mSiO_2_ (Figure 4c,d). The appearance of Si and O elements confirms the overcoating of mSiO_2_ layer on both UCNPs. The invisibility of Ho and Er elements in each kind of UCNP is due to the tiny doping amount (1 mol% of Ho or Er) compared to the host material of LiYF_4_ (74 mol% of Y). In contrast, the presence of Cu element can be attributed to the analyzed materials that were deposited on the copper substrates for examination.

### 3.3. Optical Properties

Since the PL intensity is a crucial factor for the efficiency of the FRET process between the UCNP donor and the PS acceptor, the PL spectra of both UCNPs before and after coating of mesoporous silica layer are shown in Figure 5a,b. As reported in our previous article, the PL spectra are almost the same for both UCNPs before and after removing the oleate ligand. Therefore, only the PL spectra of oleate-capped and mSiO_2_-coated UCNPs are compared in this figure. For the oleate-capped Ho^3+^-doped UCNPs, the PL spectrum presents the characteristic emission peaks of 540 nm and 650 nm (Figure 5a). With regard to the oleate-capped Er^3+^-doped nanoparticles, the PL spectrum exhibits three emission peaks around 525, 550, and 650 nm (Figure 5b). It is evident that overcoating the mesoporous silica layer on the surface of UCNPs has no prominent influence on the PL intensity. Conversely, it is beneficial for UCNPs to generate more ROS because the mSiO_2_ layer can adsorb a greater amount of photosensitizer. 

### 3.4. Loading Capacities of Photosensitizers and Fluorescence Resonance Energy Transfer

Taking advantage of the green emission and red emission function of UCNP@mSiO_2_ nanocomposites, a therapeutic nanoplatform for photodynamic therapy (PDT) is constructed by incorporating photosensitizers into the UCNP@mSiO_2_ nanohybrids. Specifically, we have chosen MC 540 and RB as the photosensitizers of Ho^3+^-doped UCNP@mSiO_2_ in PDT treatment, as they have strong UV/Vis absorptions around 400–600 nm that overlap well with the dominant green emission (540 nm) of Ho^3+^-doped UCNP materials (Appendix A). However, Ce 6 has been adopted as the photosensitizer of Er^3+^-doped UCNP@mSiO_2_ because it has a strong absorption overlapping with the red emission (650 nm) of the Er^3+^-doped UCNPs, as shown in Appendix A. Due to the efficient overlap between the emissions of UCNP@mSiO_2_ nanoparticles and the absorptions of photosensitizers, the effective FRET is expected to occur from UCNP@mSiO_2_ nanomaterials to photosensitizers. 

For the NIR-triggered PDT, the therapeutic strategy is associated with the photochemical reactions of photosensitizers to generate cytotoxic ROS to destroy nearby cancer cells. Accordingly, in order to generate a large amount of ROS for PDT, it is crucial that the effective FRET takes place between UCNPs and photosensitizers upon NIR irradiation. Directly coating the mSiO_2_ layer for loading more PS molecules on the UCNP surface is considered to facilitate more ROS generation compared to other surface modification strategy.

To perform the effective FRET process between UCNP@mSiO_2_ nanocomposites and photosensitizers, the loading capacities (%, *w*/*w*) of both mesoporous silica-coated UCNPs with their corresponding photosensitizers were determined by the UV-vis absorption spectra of standard photosensitizer solutions at different concentrations. In order to confirm that coating mSiO_2_ layer on the surface of UCNPs is a more efficient modality to produce ROS for PDT compared to other synthesized UCNPs with similar compositions, the loading capacities of oleate-capped and ligand-free UCNPs with their correlative photosensitizers were also listed in Table 1. As seen in Table 1, for the three kinds of UCNP—PS pair, denoted as Ho^3+^-doped UCNPs/MC 540, Ho^3+^-doped UCNPs/RB, and Er^3+^-doped UCNPs/Ce6, respectively, the adsorbed amounts of photosensitizers for mSiO_2_-coated UCNPs are the largest among the three different types of UCNPs. It is noteworthy that the adsorbed amount of MC 540 reaches about 30.6% (*w*/*w*) for every gram of Ho^3+^-doped UCNP@mSiO_2_. This result demonstrates that increasing the surface area of UCNPs by means of coating the UCNP surface with the mesoporous silica is a promising strategy to raise the adsorbed amounts of photosensitizers on the surface of UCNPs. 

The efficacy of FRET between UCNPs and photosensitizers could be evaluated from the quenching efficiency of PL spectra of UCNPs before and after adsorption of the photosensitizers. Based on the peak intensity of the green emission and red emission before (F_max_) and after (F_t_) adsorption of the photosensitizer molecules on the surface of UCNPs, the quenching efficiency (%) can be obtained using the equation of (F_max_ − F_t_)/F_max_. The PL spectra due to FRET between both Ho^3+^-doped UCNP@mSiO_2_ and Er^3+^-doped UCNP@mSiO_2_ with their corresponding photosensitizers are shown in Figure 6. For comparison, the PL spectra of the ligand-free type UCNPs with their corresponding photosensitizers are also displayed in this figure. For completeness of the comparison, two figures from the authors’ previous work [50] are included as shown in Figure 6a,e. As noted above, the PL intensities of both ligand-free UCNPs are almost the same as those of oleate-capped UCNPs. It is evident that effective FRET processes between UCNP@mSiO_2_ nanocomposites and their corresponding PSs can be observed from Figure 6. 

The calculated quenching efficiencies of the three different types of UCNP—PS pairs are illustrated in Figure 7. As seen in Figure 7, it is apparent that among the three different UCNP—PS pairs, the quenching efficiency resulting from the FRET of Ho^3+^-doped UCNPs to MC 540 is the greatest due to the larger spectral overlap region as shown in Appendix A. Compared to the oleate-capped UCNPs, both ligand-free and mSiO_2_-coated UCNPs have greater quenching efficiencies, which can be ascribed to the close adjacency between the FRET pairs of the UCNP donors and the PS acceptors and the larger adsorbed amount of PS molecules. The quenching efficiencies for Ho^3+^-doped UCNP@mSiO_2_ with MC 540 or rose bengal, and Er^3+^-doped UCNP@mSiO_2_ with Ce6 are ca. 76.2%, 64.6%, and 44.1%, respectively. The results shown in Figure 7 demonstrate that the three mSiO_2_-coated UCNP/PS pairs have comparable quenching efficiencies with their corresponding ligand-free UCNP/PS pairs. In spite of this, the results which will be shown below will confirm that mSiO_2_-coated UCNP/PS pairs have more pronounced effect for ROS production in PDT as compared to the ligand-free type UCNP/PS pairs. It can be attributed to that mSiO_2_-coated UCNPs have greater loading capacities of photosensitizers in comparison with ligand-free UCNPs.

### 3.5. Evaluation of ROS Generation of UCNP@mSiO_2_/PS Nanomaterials

The generation of reactive oxygen species (ROS) from the three UCNP@mSiO_2_/PS nanomaterials were evaluated using 9,10-anthracenediyl-bis(methylene) dimalonic acid (ABDA) as the chemical probe to detect the singlet oxygen ^1^O_2_. The quenching reaction between ABDA and singlet ^1^O_2_ resulted in a decrease in UV-vis absorption intensity of ABDA. Consequently, the ROS production can be well monitored by measuring the decrease in the absorption signals of the characteristic peaks at 359 nm, 378 nm and 399 nm of ABDA under NIR irradiation. The absorption changes of the three characteristic peaks for Ho^3+^-doped UCNP@mSiO_2_/MC 540, Ho^3+^-doped UCNP@mSiO_2_/RB, and Er^3+^-doped UCNP@mSiO_2_/Ce6 donor-acceptor nanoplaforms under different irradiation times of 980 nm NIR are shown in Figure 8 and Appendix A. As seen in this figure, the absorbances of the three characteristic peaks for the three donor-acceptor pairs display a significant downward trend, indicating the occurrence of effective ROS generation. It can be seen that the A_t_/A_max_ (%) curves of three characteristic peaks are almost the same. The values of A_t_/A_max_ (%) for the three donor-acceptor systems upon 980 nm laser irradiation for 60 min decrease by about 27%, 26%, and 22%, as is evident from Figure 8 and Appendix A and Table 2. It is pronounced that the Ho^3+^-doped UCNP@mSiO_2_/MC 540 donor-acceptor pair exhibited the largest ROS generation among the three donor-acceptor pairs owing to the larger spectral overlap area and the greater loading capacity of MC 540 for efficient FRET. Most importantly, as illustrated in Appendix A and Table 2, it is apparent that after overcoating the UCNP surface with the mesoporous silica layer on the UCNP surface, the generation amounts of ROS for the three donor-acceptor pairs are significantly increased compared to their corresponding oleate-capped and ligand-free type donor-acceptor pairs. Consequently, it is suggested that surface modification of UCNPs via coating an mSiO_2_ layer may be a facile and efficient method for ROS generation in PDT. 

### 3.6. In Vitro Photodynamic Therapy of UCNP@mSiO_2_/PS Nanomaterials with MCF-7 Cancer Cells

The cytotoxicity test of the three UCNP@mSiO_2_/PS nanomaterials was performed on a human breast-cancer cell line MCF-7. Figure 9a,b illustrates the viability of MCF-7 cells incubated with UCNP@mSiO_2_/PS at different concentrations without and under laser light irradiation, respectively. As shown in Figure 9a, it is evident that the cell viability is still over 75% when up to 500 μg/mL of UCNP@mSiO_2_/PS nanocomposites were incubated with the cells without NIR exposure. Conversely, if the three UCNP@mSiO_2_/PS nanocomposites were incubated with the cells at different concentrations and irradiated with NIR laser light, the cell viabilities for the three nanocomposites significantly drop to ca. 39% at the concentration of 500 μg/mL (Figure 9b), suggesting the occurrence of effectual ROS generation for PDT. Figure 10 is images of MCF-7 breast cells incubated with various UCNPs and PSs without and with NIR irradiation, respectively. The number of cells under NIR irradiation is less than that of particle-treated controls (unirradiated samples), which agreed with the results presented in Figure 9.

## 4. Conclusions

In this work, two mSiO_2_-coated LiYF_4_ UCNPs with the corresponding emission in green and red regions were prepared and used as the donors for the study of ROS generation and in vitro PDT treatment. Three PSs were employed as the acceptors to fabricate the PDT theranostic nanoplatform via the donor/acceptor FRET process. In comparison with both oleate-capped and ligand-free UCNPs, the mSiO_2_-coated UCNPs achieved a high loading capacity of 30.6% (*w*/*w*) for MC 540. The results of quenching efficiencies between both mSiO_2_-coated UCNPs and their corresponding photosensitizers indicated that the FRET efficacies were close to those of ligand-free type UCNP/PS pairs and much greater than those of oleate-capped UCNP/PS pairs due to the close proximity of the donor-acceptor pair after ligand removal. Furthermore, the quenching efficiency was more prominent for Ho^3+^-doped UCNPs as compared to Er^3+^-doped UCNPs because of the larger spectral overlap for effective FRET process between the UCNPs and the PS molecules. With regard to the three UCNP@mSiO_2_/PS nanocarriers, due to the effect of close adjacency between donors and acceptors as well as the higher loading capacities of PS molecules, the amounts of ROS generation were more prominent as compared to the ligand-free and the oleate-capped UCNP/PS pairs. In vitro cytotoxicity tests indicated that the cell viabilities for the three UCNP@mSiO_2_/PS nanocomposites dropped to ca. 39% at the concentration of 500 μg/mL, suggesting that the LiYF_4_ UCNP@mSiO_2_/PS donor-acceptor nanoplatform could be utilized as an effectual approach in photodynamic therapy.

## Data Availability

The authors confirm that the data supporting the findings of this study are available within the article and its Appendix A.

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
