# Peer review of "NIR-Triggered Generation of Reactive Oxygen Species and Photodynamic Therapy Based on Mesoporous Silica-Coated LiYF4 Upconverting Nanoparticles"

_ijms, 2022, doi:10.3390/ijms23158757_

Round 1

Reviewer 1 Report

This paper considers nanosystems for PDT based on m-SiO2-coated upconverting nanoparticles loaded with different photosensitizers. Such systems demonstrate effective ROS generation and photodynamic activity in vitro due to the high loading capacity of the nanoparticles coated with the silica shell and a close distance between the nanoparticle core and the photosensitizer molecules favoring FRET. The main advantage of such nanosystems for PDT is the NIR excitation providing a deeper light penetration compared to the conventional visible light photosensitizers.

The introduction provides a brief overview of the problem addressed based on the current references. The hybrid nanosystems obtained are fully characterized by a number of physico-chemical methods, including XRD, TEM, DLS, XPS and EDS. Their properties are studied by UV-vis and photoluminescence spectroscopy, photochemical measurements and cytotoxicity estimation. The results obtained are well presented, convincing and self-consistent with the conclusions being supported by the results.

This paper is expected to be of great interest for the readers dealing with nanocarriers for drug delivery and photosensitizers for PDT, so it can be recommended for publication in the current form with minor grammatical corrections after a careful proofreading. It would be also useful to reduce the number of figures presented in the manuscript by removing them into the supplementary materials.

Reviewer 2 Report

In this paper, the authors describe the synthesis and characterization of a complex nanosystem based on LiYF4 upconversion nanoparticles coated with mesoporous silica nanoparticles in order to encapsulate in their mesoporous photosensitizers. In this way, they obtained hybrid nanoparticles which absorbed in the NIR region (980 nm) and emitted in the green or red region to excite the photosensitizer (MC 540, RB or Ce6) by the FRET phenomenon.

Moreover, they demonstrated the capability of their nanoparticles in vitro in MCF-7 cells.

Although I feel, the authors should review the format carefully. I recommend the acceptance of this paper after minor revision.

The author should consider these comments before its publication:

- In the abstract, I recommend adding the type of cells used in this work human breast cancer cell line MCF-7, it is an important part of the paper and until the page 17 does not appear.

- Secondly, I feel that it could be clarified the presence of the scheme of the design of the nanoparticles, to make the visualization of the systems easy for the readers. And comparing the three different types of particles: i) oleate, ii) ligand-free and iii) UCNP@mSiO2.

- On the other hand, in line 215 the authors affirm “it is apparent that both mSiO2-coated UCNPs displayed excellent dispersibility in aqueous solution.” but they did not show values of potential Z for these samples. I recommend adding.

- In addition, I suggest adding in the ESI the results of ROS of oleate and ligand-free nanoparticles to demonstrate the sentence in line 374 “ In spite of this, the results which will be shown below will confirm that mSiO2-coated UCNP/PS pairs have more pronounced effect for ROS production in PDT as compared to the ligand-free type UCNP/PS pairs.” Because of the result of ligand-free type, UCNP/PS pairs are not shown in this version, only the values in Table 2.

- I propose moving Figures 8 c, d, e, and f to the ESI due to the results of the three nanosystems are similar.

- Finally, there are some mistakes in the format.

 For example:

- In Figures 2, 3and 9 do not appear the letter “a” and “b” in the images, I recommend adding in the images or changing the caption and putting “left” and “right”.

- The same appends in Figure 4 but which the four graphs.

- Moreover, in some Figures, the letters are not close to the graphs/figures. For example figure 5 A, Figure 6 b, Figure 8 a, b, c, d, e, f.

- I would added a new subsection after the table 2 to separate the in vitro results.
